# Servant leadership and employee prosocial rule-breaking: The underlying effects of psychological safety and compassion at work

**Naqib Ullah Khan** [1]*, **Muhammad Zada**[2,3]*, **Christophe Estay**[4]

1 School of Public Administration, Central South University, Changsha, China, 2 Facultad de Administración y Negocios, Universidad Autónoma de Chile, Santiago, Chile, 3 Department of Management Science and Commerce, Alhamd Islamic University, Islamabad, Pakistan, 4 FERRANDI Paris, France / University of Lyon, Jean Moulin, Magellan, France, Paris, France

* naqibkhan05@csu.edu.cn (NUK); muhammed.zada@uautonoma.cl (MZ)

## Abstract

The study intends to investigate the impact of servant leadership on pro-social rule-breaking directly and through the intervening mechanism of psychological safety. More, the study also plans to investigate whether compassion at work moderates the effect of servant leadership on psychological safety and pro-social rule-breaking and the indirect intervening effect of psychological safety between servant leadership and pro-social rule-breaking. Responses were collected from 273 frontline public servants in Pakistan. Using social information processing theory, the results revealed that servant leadership positively influences pro-social rule-breaking and psychological safety and that psychological safety influences pro-social rule-breaking. Results also revealed that psychological safety acts as an intervening mechanism in the relationship between servant leadership and pro-social rule-breaking. Moreover, compassion at work significantly moderates the relationships of servant leadership with psychological safety and pro-social rule-breaking, and that compassion at work ultimately alters the size of the intervening effect of psychological safety between the connection of servant leadership and pro-social rule-breaking.

## Introduction

Policies and rules are widely recognized as essential features of bureaucracies [1], which are used for achieving various purposes including guiding decision-making, promoting collaboration [2], and regulating, limiting, directing, and streamlining employees' behaviors [3]. However, some rules can be overly complicated, restrictive, and or even entirely redundant, which limits the employees' flexibility and discretion to contribute effectively and efficiently [4]. The existing written rules are not free from shortcomings in resolving intricacies at the workplace, and are continuously becoming outdated because of the rapidly changing technological innovations and uncertainties in the external environment. Employees particularly frontline public servants continuously break formal policies and rules that restrict rather than promote organizational efficiency [5]. An employee rule-breaking that is intended to promote organizational

**Competing interests:** The authors have declared that no competing interests exist.

efficiency and other positive outcomes is referred to as pro-social rule-breaking (PSRB) [6]. PSRB is conducted for various pro-social reasons and has numerous constructive outcomes including creating a cooperative workplace [7], improving employees' and organizational performance, efficiency, and image [6, 8], PSRB indicates shortcomings in the applied regulatory protocols, which leads to the design of better protocols [5]. Unfortunately, the existing empirical literature about PSRB is limited.

No matter whether PSRB is a constructive behavior where the intentions of the rule-breaker (employee) are positive (unselfish), however, such an employee is confused by expecting both rewards for doing beyond prescribed job responsibilities and simultaneously anticipating punishment for rule-breaking [6]. Social information processing theory (SIPT) posits that individuals evaluate the potential risk of punishment before expressing certain behaviors like PSRB [9]. A supervisor is the most immediate source of punishment because of the possession of authority in the organization, due to which employees actively evaluate the potential actions of the supervisor before engaging in PSRB [10]. Supervisors with ethical and inclusive leadership styles have significant impacts on PSRB [10, 11]. Compared to these leadership styles, servant leadership (SL) has more unique characteristics like keeping good interpersonal relationships with subordinates [12] and taking care of their emotional, ethical, spiritual, and relational needs [13]. The perceptions of the characteristics of SL may encourage subordinates that the leader will not act against them blindly and that the motives behind and importance of PSRB will be understood by their leader. Using SIPT, the subordinates of SL may assume that their PSRB will be rewarded or at least not punished. This research investigates the potential impact of SL on PSRB.

Positive managerial leadership is a potential source of changing the psychological conditions of subordinates which ultimately pushes them to act constructively [14]. Psychological safety is referred to as an employee's belief that it is safe to share ideas, questions, concerns, and even mistakes at the workplace [15]. High psychological safety encourages employees' risk-taking behavior to learn, innovate and experiment with new methods to achieve organizational goals [16]. Leadership style is one of the most important factors for stimulating or inhibiting followers' psychological safety [17], and among the characteristics of SL provides a more favorable working environment for the followers to feel safe [18], and high psychological safety is an encouraging force for violating rules for pro-social purposes [10]. The existing literature on the relationships between SL and psychological safety is scarceregarding the relationship between psychological safety and PSRB, particularly there is a lack of empirical research on how SL influences PSRB through the mediating mechanism of psychological safety. More, the antecedents and outcomes of psychological safety are not stable across contexts, therefore researchers emphasize additional studies from different samples to enhance the generalizability of the construct of psychological safety [19, 20]. This study is intended to investigate the relationships between psychological safety with SL and PSRB, and more importantly, explore the mediating impact of psychological safety on the relationship between SL and PSRB.

Moreover, the perceived compassionate support from leadership and co-workers changes employee opinions about the overall work environment, which ultimately improves his/her psychological condition [21]. Compassion at work (CAW) is an important interpersonal process in which individuals notice, feel and act to ease the miseries of others [22], and individuals with high perceptions of compassion from leadership and co-workers show high gratitudes which inspire them to reciprocate and forward compassion when they observe others adversaries [23]. Perceptions of CAW arouse feelings of acceptance, care, and happiness [24], and it is an influential mechanism for alleviating work stress and promoting psychological energy [25], which ultimately encourages cooperative and prosocial behavioral outcomes [26]. CAW is recognized as an important moderating mechanism between different antecedents and

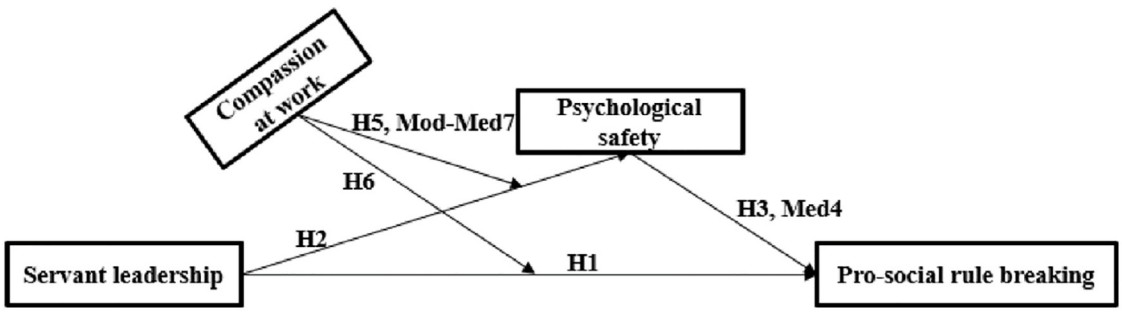

**Fig 1. Research model.**

employees' outcomes [27, 28] including the effect of leadership on followers' pro-active behaviors [29]. The employees perceived high and low CAW change their opinions about the leadership accordingly. That is the perceptions of high CAW develop followers' confidence and trust while low CAW arouses feelings of discomfort from leadership, which alters the influence of leadership accordingly. Using SIPT, this study assumes that the sizes of the impacts of SL on psychological safety and PSRB are dependent on the extent of perceived CAW and that the extent of perceived high and low CAW may influence the size of the mediating impact of psychological safety between the relationship of SL and PSRB. Based on this discussion, this research plans to test the following overall theoretical model (Fig 1):

## Hypotheses development

### Relationship of SL and PSRB

SL theory has attracted scholars' and practitioners' attention over the past four decades and this has become one of the most desirable leadership behaviors for influencing the outcomes of frontline service providers specifically during pandemic [30–32]. Greenleaf [33] has defined SL as a servant first which begins from the natural feeling that one wants to serve first. SL is known as a unique self-sacrificing leadership behavior in the interest of followers [34]. SL provides the opportunity for subordinates to share opinions [35], allows them to make independent decisions, and appreciates new ways of working and problem-solving [36]. Servant leaders are available for providing timely feedback when subordinates encounter problems at the workplace. SL keeps a close interpersonal relationship with their followers [37], and takes care of their personal and professional needs [38] which easily develops the subordinates' trust in leadership [39]. Lee, Lyubovnikova [13] systematic review concludes that SL has the characteristics to satisfy subordinates based on emotional, ethical, spiritual, and relational grounds. According to Hoch, Bommer [40], the empirical power of SL is higher than other positive leadership styles, and it has emerged as one of the most important leadership styles for producing numerous positive outcomes including improving employees' commitment [41], engagement [42], motivation and job performance [43], creativity and innovations [44], work-related well-being [45], innovative work behavior [46], citizenship behaviors [47], positive deviance [48], productive voice behavior [49] and pro-social behaviors [50].

PSRB is a distinct kind of pro-social behavior where an employee breaks formal rules but with positive intentions and constructive purposes. PSRB is somewhat opposite to destructive deviance where employees break rules and norms for promoting personal rather than organizational interests [48]. PSRB is referred to as the employees' violation of written policies and rules with motives to primarily promote the interests of the organizations [6]. The

prevailing written rules have deficiencies and are continuously becoming outdated because of the fast technological innovations and other unexpected changes in the external environment [51, 52]. Therefore, employees frequently engage in PSRB particularly on occasions when they notice that the formal rules and procedures are inflexible and outdated for their work efficiency. They break rules for various prosocial reasons, for example, to stop their due processes to share expertise for helping co-workers, to deal with issues of organizational clients (e.g. citizens) in off timings, and to facilitate other stakeholders of organizations [7, 53]. Bureaucracy is usually rule overburdened and the frontline public servants have first-hand information regarding the issues at the workplace [4], therefore they are highly likely to break formal rules which restrict rather than promote their service efficiency [5, 51]. PSRB has received the attention of scholars and practitioners for improving employees and organizational outcomes. However, the existing literature is not sufficient for understanding the determining factors of PSRB.

SIPT poses that the external environment influences individuals' attitudes and behaviors, that is individuals engage in those behaviors which are perceived to be rewarded or at least not punished by the external factors [9]. Leadership style is one of the most important situational factors for influencing employees' behaviors because of the power and position held in organizations [14, 54], and employees evaluate the potential response of supervisors before participating in certain behaviors like PSRB [10]. Although, employees' engagement in PSRB is selfless and ultimately improves the organizational outcomes [5, 7], however, employees may expect both rewards for contributing beyond the job description and punishment for rule-breaking [6]. Therefore, they may change the frequency of participation in PSRB based on the expected reward or punishment [6]. When employees expect that the supervisor may understand the motives behind the benefits of PSRB, then they may increase participation in PSRB, and vice versa [10]. Based on the unique followers-centered characteristics of servant leaders, we assume that the perceived attributes of SL may positively influence PSRB. This discussion leads toward the establishment of the following hypothesis.

**H1**: SL is positively associated with PSRB

## The mediating effect of psychological safety

Psychological safety is employees' confidence that their personal and professional interests will not harm by expressing themselves at work [55]. An employee is feeling psychologically safe when he/she perceives that there are no negative consequences from leadership, co-workers, and overall work context for expressing ideas, questions, concerns, and even mistakes [15]. PSRB is acting against the formal rules and procedures therefore it has the attached risk to be punished by the supervisor [6]. It is commonly understood that employees will evaluate the attached risk of punishment from their supervisors before breaking rules even for pro-social reasons. It is expected that employees will act against formal rules only when they are feeling psychologically safe and confident that such rule-breaking will not cause negative consequences from their leadership [10]. Positive leadership is one of the most important factors in encouraging employees' psychological safety [10]. Employees' psychological safety prospers in a work environment where they receive leadership support for accomplishing their personal and professional needs [56]. As mentioned earlier SL is a unique follower's centered leadership approach that prioritizes caring for the subordinates' emotional, ethical, spiritual, and relational needs, provides them accessibility through keeping close interpersonal relationships, and empowers them to act independently to use new ways for accomplishing tasks. The perceived characteristics of SL are likely to encourage subordinates to indulge in useful but risky

rule-breaking, due to altering the psychological conditions of them that the supervisor is accessible who may understand the essentiality of such rule-breaking. Limited research has noted that SL with servant characteristics is a driving force for promoting followers' psychological safety [57] and that high psychological safety encourages PSRB [10], and the literature supports that the antecedents and outcomes of psychological safety are not consistent, therefore scholars emphasize additional studies in different contexts using different samples for improving the generalizability of the construct [17, 19], particularly there is lack of any empirical research evidence on how SL promotes PSRB via the mediating mechanism of psychological safety. Using SIPT, this study assumes that the perceptions of the subordinates about the characteristics of SL may reduce their fear of punishment, improves their expectations for reward, which consequently enhance their psychological safety, and which ultimately encourage them to participate in PSRB. Based on this discussion, we hypothesize the following hypotheses:

**H2**: SL is positively associated with psychological safety

**H3**: Psychological safety is positively associated with PSRB

**H4**: Psychological safety mediates the association of SL with PSRB

## The moderating effect of compassion at work

Compassion at work (CAW) is referred to as the sensitivity of an individual to feel and lessen the sufferings of others [58]. Compassionate feelings activate employees' empathy, love, and care for alleviating the miseries of others [26]. Compassion is highly important in bureaucracy where frontline public employees bear sufferings because of working in complex and fragile settings [4]. Compassion is known as co-suffering, individuals with high perceptions of compassion from leadership and peers show high gratitude and such individuals reciprocate and forward compassion when they observe someone else's sufferings [23]. High workplace compassion alleviates work stress and anxieties by arousing positive emotions [59] and engenders positive interpersonal relationships [22]. Perceived CAW is an external situational factor that alters individuals' behavioral outcomes [60]. According to SIPT, individuals evaluate the clues provided by the external environment and consequently adjust their attitudes and behaviors accordingly. Perceptions of compassion stimulate positive feelings and emotions and stimulate the perceptions of self-worth and psychological freedom [25]. CAW is referred to as the perceptions of employees that the leadership and co-workers and overall job environment are compassionate [58]. Earlier studies note that perceived CAW is an important external factor that alters the influence of leadership on psychological safety [61]. This association is likely to be stronger when the followers perceive compassion in the workplace. That is evaluating the perceived high and low CAW may alter followers' opinions about the leadership, which ultimately may influence their psychological safety. Using the propositions of SIPT, this study assumes that the size of the impact of SL on psychological safety is dependent on the extent of perceived CAW.

Research notes that the perceptions of high CAW are an important mechanism for promoting prosocial behavioral outcomes [62]. High perceived compassion improves individuals' confidence and trust in leadership which improves their extra activism [22, 63]. According to SIPT, employees may break rules more actively when they perceive CAW because the perceptions of high compassionate support may develop their confidence in SL. More, rule-breaking causes employees psychological stress as it has an associated risk of harming their personal and professional interests, while SL satisfies followers' interests. Therefore, high CAW may improve employees' acceptance of SL more confidently that their personal and professional

interests are protected, which encourages them to more actively engage in PSRB. In contrast, if employees don't feel CAW, then this may create a trust deficit in employees from leadership which restricts them from actively participating in PSRB. CAW is found as an important moderating factor between the association of forgiving behavior and stress alleviation [64], and the extent of CAW is found as a moderating factor between the effect of leadership style and followers' proactive behavior [61]. SIPT and the relevant literature encourage us to assume that the perceptions of high CAW may influence the size of the effect of SL on PSRB.

Moreover, as we have assumed above psychological safety mediates the effect of SL on PSRB. The perceptions of high CAW may moderate the mediating effect of psychological safety between the association of SL and PSRB, such as, high CAW may improve employees' confidence in SL which may alter the effect of SL on psychological safety, and which ultimately may moderate the size of the mediating effect of psychological safety. That is the mediating effect of psychological safety is contingent upon the extent of perceived CAW. Based on this discussion, the following hypotheses are presented for testing.

**H5**: CAW moderates the association between SL and psychological safety

**H6**: CAW moderates the association between SL and PSRB

**H7**: CAW moderates the mediating effect of psychological safety between the association of SL and PSRB, (that is the size of the mediating effect changes with the extent of the perceptions of high and low CAW).

## Methods

### Sample and procedures

The sample of the study is the frontline public servants who are actively engaged at the interface between government and citizens in the provision of welfare services under the umbrella of the "Ehsaas (care) Programme" across Pakistan. This program was launched at the beginning of 2019 by the government of Pakistan across various districts focusing on uplifting marginalized people, reducing inequality by empowering women, and balancing regional socioeconomic development. The Ehsaas Programme continues to implement several social safety and poverty alleviation schemes including "The Ehsaas emergency cash program", "Ehsaas Kafalat (livelihood) program", "Panahgah (shelter houses)", "Langarkhane (community kitchens)", and "Ehsaas public health program". These schemes continue to facilitate thousands of poor citizens daily through the provision of various public welfare services including the provision of emergency cash to the poor specifically women; provision of temporary shelters for the homeless and passengers; provision of food to the poor and daily wage earners; and the provision of public health facilities to the citizens. The surveys were conducted with the approval of the Ministry of Commerce and Industry, social safety division, and districts administrations using convenient sampling procedures. The initial plan was to collect data using two-stage descriptive paper and pencil-based surveys from around 300 frontline public servants working under the immediate supervision of public officials in the above-mentioned public welfare schemes. The motivation behind using two-stage surveys was to minimize the risk of common method bias (CMB) which is caused by data collection using self-report measures and lengthy questionnaires [65]. Two sets of questionnaires and one set of response request letters were prepared, where the first set of questionnaires were measuring servant leadership, control variables, and questions for identifying the personal identities of the respondents, and the second set of questionnaires were having questions for measuring PSRB, psychological safety, and CAW, and the response request letters were having information regarding the purpose of the

surveys, ensuring anonymity of responses and seeking approval for voluntary participation. During the first stage of the surveys, the research team (including one Ph.D. student and two assistant professors from the discipline of management science) conducted survey of 315 employees. All the respondents signed the response request letters and answered the first set of questionnaires on the spot. However, we have selected only 296 questionnaires for analysis because these were validly and accurately answered and the rest were rejected based on providing inaccurate and missed information. The second stage survey was conducted after around one month, where we accessed only 296 respondents who have accurately filled out the first set of questionnaires. These particular respondents were accessed using their identities received on the first set of questionnaires. We have again emphasized the confidentiality and anonymity of the responses because we were concerned that the respondents may report the self-report measures of PSRB, psychological safety, and CAW in a socially desirable manner. During this stage, only 273 respondents provided valid answers among the 296 respondents who accurately filled out the first stage survey. Responses to the first and second-stage surveys were matched using the personal identities of the respondents. Thus 273 questionnaires were finally selected for analysis. The total response rate of the study becomes 86% based on the initially selected sample of 315 respondents. The demographics of the sampled respondents were as: 81% were males; the average age of the respondents was 29 years; 68% of respondents were having graduate-level education; 65% of respondents were performing administrative responsibilities; and the average experience of the respondents was 2.71 years.

## Measures

The constructs of the study were measured using instruments adopted from earlier literature. Responses were received on a five-point Likert scale ranging between 1 for strongly disagree to 5 for strongly agree. SL was measured seven items instrument developed by Liden, Wayne [66], and the Cronbach's alpha (α) reliability of the instrument was 0.92. Psychological safety was measured using a five items instrument developed by Liang, Farh [67], alpha reliability of the instrument was 0.89. PSRB was measured using a thirteen items instrument recommended by Dahling, Chau [7], alpha value of the instrument was 0.94. CAW was measured using a three items instrument developed byLilius, Worline [24]. This instrument measures three dimensions of compassion, that is compassion from supervisors, co-workers, and the overall job environment. One item of the instrument is expressed as: "I could feel compassion at work from my supervisors", alpha reliability value of the instrument was 0.85. Gender, age, experience, education level, and job position may influence PSRB, therefore these variables were included as controls [6, 7].

## Data analysis

Data were analyzed using SPSS, Amos, and PROCESS Macro. Data analysis was performed such as first confirmatory analysis (CFA) for confirming the factor structure of observed variables (measured items), CFA confirming whether the observed variables suitably underlying the unobserved hypothesized latent constructs [68]. The hypotheses were verified using mediation model-4 and moderation model-8 [69].

## Results

### Confirmatory factor analysis

CFA was applied for confirming whether the four-factor model (SL, PSRB, psychological safety, and CAW) fits the data. The results revealed that the four-factor model has fitted well to

**Table 1. Descriptive statistics, correlations, and constructs reliability and validity.**

| Variables | CR | AVE | Mean±SD | 1 | 2 | 3 | 4 |
|-----------|-----|-----|---------|-----|-----|-----|-----|
| 1. SL | 0.92 | 0.62 | 3.20±.35 | *0.78* | | | |
| 2. PSRB | 0.91 | 0.64 | 2.79±.25 | 0.48** | *0.76* | | |
| 3. PS | 0.88 | 0.61 | 3.53±.36 | 0.55** | 0.52** | *0.71* | |
| 4. CAW | 0.90 | 0.58 | 2.43±.29 | 0.63** | 0.26** | 0.45** | *0.74* |

Note (s): N = 273, SL = Servant leadership, PSRB = Pro-social rule breaking, PS = Psychological safety, CAW = Compassion at work, CR = Composite reliability,

AVE = Average variance extracted, Square-root of AVE are on diagonals, Correlations are off the diagonals

the data $(\chi2 = 435.72, df = 202, \chi2/df = 2.16, p < .001, CFI = .93, TLI = .91, SRMR = .04, RMSEA = .05)$. Common method bias (CMB) was verified through Harman's single factor test, as the value of the highest emerging factor was 29.41%, as the value is below the critical point of 50%, which indicates that the self-reported data has no issue of CMB [65, 70]. Data of all the four latent constructs of SL, PSRB, psychological safety, and CAW were also tested for evaluating the composite reliabilities, convergent, and discriminant validity. The values of composite reliability (CR) for all constructs were greater than 0.80, which is an indication of excellent internal consistency [71]. The convergent validity of the constructs was verified through average variance extracted (AVE). As indicated, the values of AVE are greater than 0.50, which confirms the convergent validity for all constructs. The discriminant validity of the constructs was verified using Cronbach and Meehl [72] approach. As indicated in Table 1, the values of the square root of AVE are greater than the correlations among all the constructs, which infers that the data has no issues of discriminant validity [73].

## Hypotheses testing

**Direct and indirect (mediating) results.** Hypotheses of the study were analyzed using PROCESS Macro SPSS, developed by Hayes [69]. The direct and mediating effects of the study were extracted using mediation model 4. The findings of the hypotheses tests are shown below in Table 2. The findings indicated that SL is significantly positively related to PSRB (β = 0.83, p < .01) and psychological safety (β = 0.52, p<0.01), supporting H1 and H2. Psychological safety was also positively related to PSRB (β = 0.62, p<0.01, supporting H3. Mediation model-4 was applied at 5000 Bootstraps for extracting the results of the mediating effect of psychological safety between the relationship of SL and PSRB. As indicated the indirect (mediating) effect of psychological safety between the association of SL and PSRB was (β = 0.32, p<0.01),

**Table 2. Hypotheses results.**

| Paths | Products of coefficients | | | | BootULCI/95% | | H |
|-------|--------|------|------|------|------|------|------|
| | Effect | se | T | p | LLCI | ULCI | |
| SL → PSRB | 0.83 | 0.13 | 6.64 | < .001 | 0.59 | 1.08 | H1 |
| SL → PS | 0.52 | 0.05 | 9.77 | < .001 | 0.41 | 0.62 | H2 |
| PS (M) → PSRB | 0.62 | 0.12 | 4.98 | < .001 | 0.37 | 0.86 | H3 |
| SL_x_PS → PSRB | 0.32 | 0.06 | 5.33 | < .001 | 0.22 | 0.46 | H4 |
| SL_x_CAW (W) → PS | 0.45 | 0.21 | 2.12 | < .01 | 0.03 | 0.86 | H5 |
| SL_x_CAW → PSRB | 0.42 | 0.19 | 2.17 | < .01 | 0.04 | 0.80 | H6 |

Note: N = 273, N represents the number of respondents, SL = Servant leadership, PS = Psychological safety, PSRB = Pro-social rule breaking, M = Mediating variable,

W = Moderating variable, x = interaction term, effects were extracted at 5000 Boostraps and 95% confidence intervals.

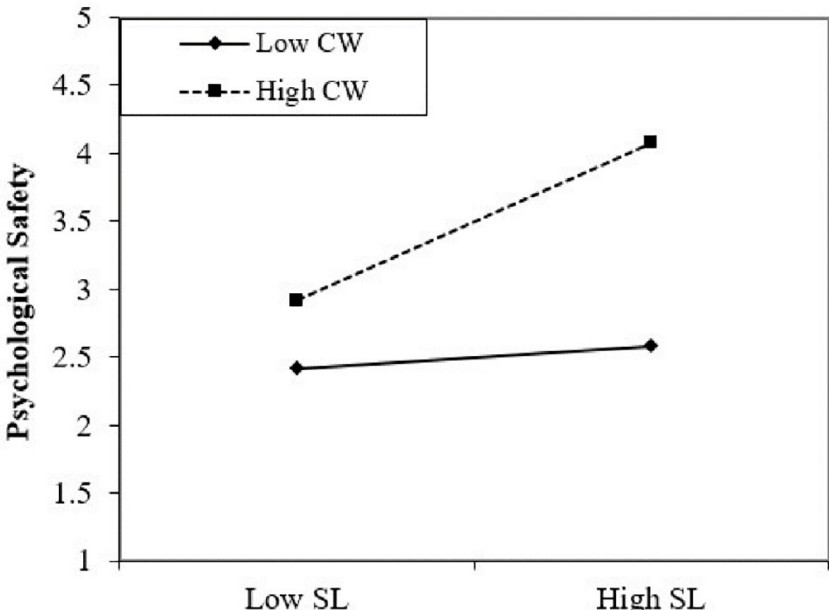

**Fig 2. Conditional effect of CAW between the relationship of SL and psychological safety.**

as shown the lower and upper extremes do not inclose 0 0[.22, 0.46] at 95% CI, which infers that the indirect effect is significantly different from 0, which signals that the mediating effect is significant, supporting H4. Thus, the regression results received support for H1, 2, 3, and 4.

**Moderating and moderated mediating results.** The moderating and moderated mediating effects of the study were extracted using moderating model-8, proposed by Hayes [69]. As indicated in Table 2, the moderating effect of CAW on the relationship between SL and psychological safety is ($\beta$ = 0.45, p<0.05), and the moderating effect of CAW on the association of SL and PSRB is ($\beta$ = 0.42, p<0.05). As shown in Table 2, 0 does not fall between the lower and upper bounds in both of these cases, which means that CAW significantly moderates the effect of SL on psychological safety and the effect of SL on PSRB as well, supporting H5 and H6. Both these moderating effects of CAW are shown in Figs 2 and 3, at 1SD below and 1SD above from the mean CAW. Moreover, the moderated mediation effect was extracted using moderation model-8 at 5000 Bootstraps and 95% CI. The findings expressed that the moderated mediation effect was insignificant ($\beta$ = 0.07, SE = 0.03, at 95% [-0.02, 0.16] at low CAW (−1SD), while the effect was significant ($\beta$ = .32, SE = .09, at 95% [.17, .53] at high CAW (+1SD). The index of moderated mediation was also significant at .21, SE = .07, at 95% [0.10, 0.38], this infers that CAW significantly moderates the mediating effect of psychological safety between the relationship of SL and PSRB [69], supporting H7. The moderating analyses received support for all hypotheses H5,6, and 7.

## Results discussion

This study intended to examine the impact of SL on PSRB. Moreover, the study also aimed to investigate whether psychological safety intervenes in the relationship between SL and PSRB and whether CAW moderates the relationships between SL with psychological safety and PSRB and the intervening path of psychological safety between the relationship of SL to PSRB. Findings showed that SL is significantly positively related to PSRB and the perceptions of SL inspire engagement in PSRB behavior. The findings of our study confirm the previous

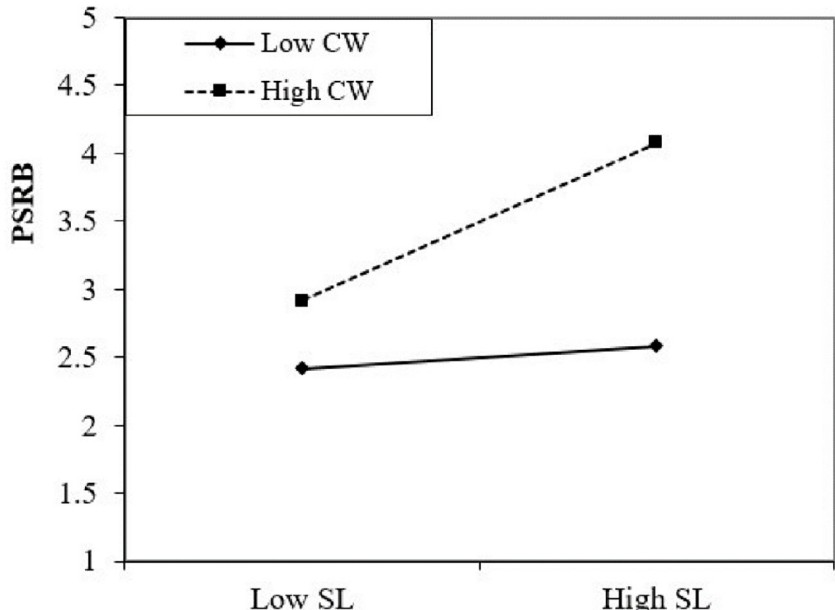

**Fig 3. Conditional effect of CAW on the relationship between SL and PS.**

literature related to other style of leadership behaviors such as ethical and inclusive leadership behaviors promote PSRB as well [10, 11]. SL is considered a positive leadership behavior that has both ethical and inclusive tendencies, and despite that, SL has uniqueness as it is the followers' enabling, accepting, respecting, altruistic, and self-sacrificing leadership behavior that satisfies followers' emotional, ethical, spiritual, and relational needs [13], and PSRB is also selfless other-oriented, therefore it is likely that the perceived experiences of SL encourage PSRB, the relationship between SL and PSRB can be observed in the light of SIPT [9]. For example when followers perceive SL then they may make the sense that their PSRB might be appreciated rather than criticized, because of the interpersonal interaction and high followers' care and support. Moreover, SL is an active aspect of leadership in the specific Islamic religious context of Pakistan somehow inspired by the teachings of Islam. Around 97% population of Pakistan is Muslim and the Prophet of Islam emphasized in his teachings that I am the servant transcended for serving humanity. Teachings of Islam emphasize serving others through financial, physical, and psychological means which are some of the unique features of SL. Moreover, SL is found as one of the most commonly practiced leadership behavior in the organizational sphere of Pakistan, and some earlier studies have noted the effects of SL on other employees' outcomes including promoting productive voice behavior [74] and demoting deviant behavior in the public sector context of Pakistan [75], which improves the likelihood that SL may promote the beneficial but risky PSRB behavior as well. Moreover, our study has noted the significant impact of SL on psychological safety. This means that perceived SL improves employees' psychological freedom for sharing ideas, issues, and mistakes, and seeking behavior for finding solutions to resolving problems. This finding is in line with some earlier studies from other contexts [56], and positive leadership behavior is noted as a determinant of inspiring psychological safety in Pakistan [76]. This study has also noted that psychological safety significantly encourages PSRB in the public service context of Pakistan. That is an increase and decrease in PSRB is dependent on the psychological safety of an employee which is received from leadership and co-workers. An earlier study has also revealed that psychological safety is

a determinant of PSRB [10]. This finding encourages us that the construct of psychological safety is stable in association with promoting PSRB which improves the generalizability of the construct towards PSRB. Moreover, the testing of the mediation hypothesis of our study revealed that SL not only directly influences PSRB but through the intervening path of psychological safety as well. Earlier studies note that psychological safety is an important underlying mechanism between leadership and followers' behavioral outcomes [10, 17, 19, 20].

Compassion is an important characteristic of people who have tendencies toward religion and spirituality [58]. Because religious texts emphasize that caring, sharing, and serving the miseries of others will give them both seen and divine rewards. Attitudes and tendencies toward compassion and co-suffering can be observed in the prevalence of many globally recognized charitable and welfare organizations such as "The Edhi Foundation" [75]. Compassion by definition is someone feeling the personal and professional worries of others which encourages to alleviate such sufferings. Compassion in Pakistan is continuously emphasized through speeches and actions by top leadership particularly the speeches at the prime ministership level. The government has recently diverted public money for running several public welfare programs (e.g. the "Ehsaas Program"). The findings of the study revealed that the perceptions of CAW moderates the size of SL on psychological safety and PSRB. Using the light of SIPT, this is perhaps because of the follower's firm belief in SL more than usual. Moreover, the construct of compassion has three dimensions, that is compassion from the overall job environment, co-workers, and leadership [24]. Studies note that SL has a distinction for promoting compassion in the work environment [45]. Thus the experiences of CAW are highly likely to change the extent of employees' perceptions of SL, which consequently may influence psychological safety and PSRB accordingly. The findings confirmed that the perceptions of CAW affect the size of the relationships of SL with psychological safety and PSRB. That is the sizes of the effects are high when the perceptions of CAW are high, and the sizes of the effects are low when perceived CAW is low, for both psychological safety and PSRB. Moreover, the moderated mediation analysis revealed that CAW influences the size of the mediating effect of psychological safety between the relationship of SL to PSRB. That is the size of the mediating effect changes according to the changes in the perceptions of CAW. Earlier studies note that CAW works as a moderating mechanism between different antecedents and employee outcomes [27, 29] including leadership behavior and followers' pro-active behavior [61].

### Significance of the study

This research contributes to the existing literature in the following ways. For example, the study found that SL is an important external factor that is significantly related to PSRB. This study also found that SL is related to psychological safety, and that psychological safety related is to PSRB, the findings were received from a unique sample of frontline public servants in the context of Pakistan, which stabilizes the earlier limited empirical findings between the relationships of SL and psychological safety [56, 77], and between psychological safety and PSRB [10]. This study also contributes that SL not only directly promotes PSRB but also engendering psychological safety. More, this study found that CAW works as a moderating mechanism between the relationship of SL with psychological safety and PSRB as well. This study also found that CAW significantly moderates the indirect effect of SL on PSRB, via psychological safety.

Based on the empirical findings of the study, we recommend the following practical implications. The occurrence of PSRB is a red signal for policymakers and other stakeholders of organizations that the policy and rule system is non complied, ineffective, and inefficient [5]. PSRB is related to promoting cooperation among employees and it is related to numerous

positive employee and organizational outcomes including commitment, engagement, performance, efficiency, and image building [6, 8]. Employees with high tendencies towards PSRB are considered passionate workers as they confront criticism for unselfish prosocial reasons to satisfy organizational stakeholders. This study informs policymakers and leaders that PSRB is a good practice that should be rewarded rather than punished. Practitioners are strongly recommended to differentiate between the constructive and destructive motives of employees before applying punishment for rule-breaking. Public sector hiring authorities are encouraged to include empathy, consciousness, risk-taking propensity, and proactiveness in the criteria of selection of particularly the frontline public servants, as employees with these attributes will be highly likely to work independently and to show PSRB [6].

This study encourages leaders to create a safe psychological work environment where employees are more expressive of their questions and opinions [15] because high psychological safety will lead employees to participate in risky but essential behaviors like PSRB. More, we emphasize that organizational managers should practice SL more often than usual because the attributes of SL may encourage employees to express themselves and ultimately participate in PSRB at the workplace. Human resource departments (HRDs) may promote SL by rewarding PSRB and by hiring public managers who have attributes for caring for the emotional, ethical, spiritual, and relational needs of others [13]. Moreover, the suffering of an individual is the suffering of the whole organization and compassion is a useful tool for alleviating individuals' sufferings. A compassionate work environment is good for organizational effectiveness and image building. CAW is a contextual mechanism and in the presence of high CAW, the followers' trust and confidence are getting stronger in leadership, which ultimately moderates their psychological safety and PSRB. We recommend public HRDs and leadership admire employees to show empathy, love, and care for each other because these attributes may contribute to the promotion of compassion, which is useful for alleviating suffering in the workplace.

## Limitations and future research directions

Data from the study were collected using a self-reporting descriptive survey procedure. The strength of using self-reporting measures is that it facilitates collecting a large number of responses in a short time and allows respondents to share their own experiences and perceptions rather than inferring based on observations of others' attitudes and behaviors. Self-reporting is suitable for measuring respondents' subjective perceptions of SL, psychological safety, PSRB, and CAW. However, this method has limitations as well, because self-reporting of PSRB, psychological safety, and CAW may raise the possibility that the respondents portray their own experiences in such a socially desirable manner. However, the risk of social desirability bias was minimized by emphasizing the provision of confidentiality through oral communications and response request letters. The second limitation of the surveys is the collection of responses from the same source (public servants only), which sometimes poses the risk of common method bias (CMB), however, this risk was reduced by collecting data at two different time intervals. Moreover, the findings from Harman's statistics revealed that there is a lack of CMB in the self-reported data. This research directs future work to use data from different sources for measuring PSRB. For example to take responses from both the leadership and subordinates for the measurement of PSRB. Researchers may measure the variables of the study at three different time points such as: measuring SL at time 1; psychological safety and CAW at time 2; and PSRB at time 3. A mixed-method approach may be applied to relook at the associations among the variables of the study, and the study can be repeated in different contexts using different samples for improving generalizability among the relationships of the variables. The traditional organizational policy and rule systems have become more complex for service

efficiency because of the implementation of covid-19 related standard operating procedures, and the CAW has received increasing attention for alleviating employees' stress and for improving their psychological well-being, and SL has received special attention for inspiring followers attitudinal and behavioral outcomes. Therefore researchers are encouraged to re-examine the relationships among the variables of the study on data collected during crises like the covid-19 pandemic. Researchers demonstrate that individuals break rules differently for different pro-social reasons [7], therefore we encourage investigating the impact of SL on individual dimensions of PSRB for understanding the phenomenon more deeply. Moreover, further studies may be conducted for investigating the impacts of other positive leadership behaviors such as authentic leadership in association with PSRB.

## Conclusion

The study aimed to investigate the relationship between servant leadership and pro-social rule-breaking. More, the study also aimed to investigate the intervening mechanism of psychological safety between the relationship of servant leadership and pro-social rule-breaking. Moreover, we also focused to examine the moderating effect of compassion at work among the relationships of servant leadership with psychological safety and pro-social rule-breaking, and on the intervening effect of psychological safety from the link of servant leadership and pro-social rule-breaking. Data were collected from 273 frontline public servants in Pakistan. The findings concluded that servant leadership promotes psychological safety and pro-social rule-breaking and that psychological safety significantly intervenes in the relationship between servant leadership and pro-social rule-breaking. That is servant leadership engenders psychological safety which onwards enhances pro-social rule-breaking. Moreover, the findings also concluded that compassion at work moderates the size of the effect of servant leadership on psychological safety and pro-social rule-breaking, and that compassion at work significantly moderates the intervening effect of psychological safety between the relationship of servant leadership and pro-social rule-breaking.

## Ethical disclosure

Institutional review board statement is in line with the Helsinki Declaration of 1964 and its later amendments to the best of our knowledge, all of the research procedures were performed within ethical standards. Formal approval was obtained from competent authorities of the organizations that participated in the study.

## Supporting information

**S1 File.**
(DOCX)

## Author Contributions

**Conceptualization:** Naqib Ullah Khan, Muhammad Zada, Christophe Estay.

**Data curation:** Naqib Ullah Khan, Muhammad Zada, Christophe Estay.

**Formal analysis:** Naqib Ullah Khan, Christophe Estay.

**Funding acquisition:** Christophe Estay.

**Investigation:** Naqib Ullah Khan, Muhammad Zada, Christophe Estay.

**Methodology:** Naqib Ullah Khan, Christophe Estay.

**Project administration:** Naqib Ullah Khan.

**Software:** Naqib Ullah Khan.

**Validation:** Muhammad Zada, Christophe Estay.

**Writing – original draft:** Naqib Ullah Khan.

**Writing – review & editing:** Christophe Estay.

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
