## [Decision Letter · Decision Letter 0]

20 Jul 2022

PONE-D-22-15473Servant Leadership and Employee Prosocial Rule-Breaking: The Underlying Effects of Psychological Safety and Compassion at WorkPLOS ONE

Dear Dr. Khan,

Thank you for submitting your manuscript to PLOS ONE. After careful consideration, we feel that it has merit but does not fully meet PLOS ONE’s publication criteria as it currently stands. Therefore, we invite you to submit a revised version of the manuscript that addresses the points raised during the review process. Editor comment. I was able to find one expert reviewer to comment on your manuscript. As you can see, the referee is positive overall and has some useful suggestions. I suggest you preparing a revision including a cover letter that contains point-by-point replies. I am looking forward to see a revision of you work soon. Please submit your revised manuscript by Sep 03 2022 11:59PM. If you will need more time than this to complete your revisions, please reply to this message or contact the journal office at plosone@plos.org. Please include the following items when submitting your revised manuscript:A rebuttal letter that responds to each point raised by the academic editor and reviewer(s). You should upload this letter as a separate file labeled 'Response to Reviewers'.A marked-up copy of your manuscript that highlights changes made to the original version. You should upload this as a separate file labeled 'Revised Manuscript with Track Changes'.An unmarked version of your revised paper without tracked changes. You should upload this as a separate file labeled 'Manuscript'.

We look forward to receiving your revised manuscript.

Kind regards,

Michael B. Steinborn, PhD

Section Editor

PLOS ONE

Journal Requirements:

3. PLOS ONE does not copy edit accepted manuscripts (https://journals.plos.org/plosone/s/criteria-for-publication#loc-5). To that effect, please ensure that your submission is free of typos and grammatical errors.

Reviewers' comments:

Reviewer's Responses to Questions

**Comments to the Author**

1. Is the manuscript technically sound, and do the data support the conclusions?

Reviewer #1: Partly

2. Has the statistical analysis been performed appropriately and rigorously? 

Reviewer #1: I Don't Know

3. Have the authors made all data underlying the findings in their manuscript fully available?

Reviewer #1: No

4. Is the manuscript presented in an intelligible fashion and written in standard English?

Reviewer #1: No

5. Review Comments to the Author

Reviewer #1: This study investigates the relationship of servant leadership, prosocial rule-breaking, psychological safety and compassion at work. They find that servant leadership significantly influences prosocial rule-breaking and psychological safety. Moreover, psychological safety affects prosocial rule-breaking and psychological safety moderates the effect of servant leadership and prosocial rule-breaking. Finally, compassion at work moderates the relationships of servant leadership, psychological safety and prosocial rule-breaking.

The study is interesting and methodologically seemingly well done. However, I have several concerns which are described in the following:

1. In the Introduction the authors describe positive rule-breaking as always unselfish (l.57). However, this is not necessarily the case. Positive rule-breaking only implies a positive consequence for others or e.g., an organization. In turn, this does not exclude profits for the individual. Actually, in most cases a positive consequence for the organization might also lead to positive consequences for the individual. For instance, the company increases revenues, so the individual gets a higher salary.

2. There is a big overlap between the Introduction and Hypotheses development. The authors might consider to check for such overlaps in order to shorten both sections.

3. Please provide a power analysis for your sample.

4. It is not really clear for me how the exact procedure of the study looks. The authors should provide more detailed information here, e.g., in which order did participants fill out the questionnaires.

5. The authors might consider to upload anonymized raw data to the open science framework (or a comparable provider) in order to enhance transparency.

6. I am not an expert regarding this kind of analyses, so I will not go into detail regarding the data analysis. However, there are some formal issues. E.g., the value for beta is missing in line 415. Also regarding the statistics in Table 2: p = .00 is very rarely the correct expression. Usually, you indicate p-values to three decimal places and would state p < .001.

7. In general, the whole manuscript requires further attention regarding details like spelling and punctuation. There are several mistakes (e.g., l. 376,386,421,610-612 to just name a few).

8. The same applies to the reference section. Several references are not cited correctly (e.g., missing page numbers, wrong capitalization,…). Doi numbers are missing for all references as well.

To sum up, I think this manuscript can be a good fit for Plos One. However, the authors definitely need to address the issues above, before I can recommend publication.

6. PLOS authors have the option to publish the peer review history of their article (what does this mean?). If published, this will include your full peer review and any attached files.

Reviewer #1: No

---

## [Author Response · Author response to Decision Letter 0]

28 Jan 2023

Cover Letter of Revisions 

Title: Servant Leadership and Employee Prosocial Rule-Breaking: The Underlying Effects of Psychological Safety and Compassion at Work

We are grateful to the editor for his/her interest in our work and thankful to the editor for providing their valuable comments. Comments were beneficial and constructive that had the potential of greatly increasing the contribution of this research. The paper has been revised to incorporate the editor comments. In the revised paper, the changes (additions) arising from the review are highlighted in yellow. Also, some additional changes made are incorporated to improve the paper. The following provides an explanation of the author’s response to the editor comments.

S.no Comments Responses 

1 Please ensure that you refer to Table 1 in your text as, if accepted, production will need this reference to link the reader to the Table.

R1: Dear Sir/Madam, the reference to table 1 is cited now at line 342 in the “Confirmatory Factor Analysis” section. This change is marked yellow there, please confirm. Thanks

2 We note that your author list was updated during the revision process. Please complete our Authorship Change Form by following this link: http://journals.plos.org/plosone/s/file?id=13d0/plos-one-change-to-authorship-form.docx If you are adding or removing more than 2 authors, you can complete multiple forms and submit as many forms as needed to reflect the updates. Please return the form(s) as an attachment by emailing plosone@plos.org or by uploading it as a submission file labeled with the file type ‘Other’. Please note that if your manuscript is accepted, we will not be able to complete the publication process without the completed form.

R2: Dear Sir/Madam, we have previously added Prof Peng Zhongyi as a co-author because he the PhD tutor of the first and main author of the manuscript, his name is withdrawn from the manuscript now because he has not significant contribution in the manuscript due to which he himself showed willingness to withdraw his name. More, we have added two co-authors and Profs Heesup Han and Chang Tang because they both have significant contributions to the manuscript particularly during preparing the reviews on the manuscript. Moreover, we have completed the journal authorship change form which is submitted separately. Please look at the authorship change form. Thanks

3 We note your current Data Availability statement is:

- "Yes - all data are fully available without restriction"

- "Data is available on request"

- "Tick here if the URLs/accession numbers/DOIs will be available only after acceptance of the manuscript for publication so that we can ensure their inclusion before publication." 

R3: Dear Sir/Madam, data is confidential and the details of confidentiality is mentioned in the “sample and procedure” section of the manuscript. But access to data can be provided only on a reasonable request by the editor from the corresponding authors. Please see at the detail answer given to the below question b. Thanks

PLOS journals require authors to make all data necessary to replicate their study’s findings publicly available without restriction at the time of publication. When specific legal or ethical restrictions prohibit public sharing of a data set, authors must indicate how others may obtain access to the data. For more information, please see https://journals.plos.org/plosone/s/data-availability.

a If there are no restrictions, please upload the minimal anonymized data set necessary to replicate your study findings to a stable, public repository and provide us with the relevant URLs, DOIs, or accession numbers. For a list of recommended repositories, please see https://journals.plos.org/plosone/s/recommended-repositories. You also have the option of uploading the data as Supporting Information files, but we would recommend depositing data directly to a data repository if possible.

R4: Dear Sir/Madam, we can public the data on appropriate request by the editor, the data availability statement is added after the conclusion section at the end of the manuscript. Thanks 

b If there are ethical or legal restrictions on sharing a de-identified data set, please explain them in detail (e.g., data contain potentially identifying or sensitive patient information, data are owned by a third-party organization, etc.) and who has imposed them (e.g., a Research Ethics Committee or Institutional Review Board, etc.). Please also provide non-author contact information* for a data access committee, ethics committee, or other institutional body to which data requests may be sent. 

R5: Dear Sir/Madam, during attracting respondents for participating in the surveys, we have ensured them the confidentiality or anonymity of their responses through signing the response request letters, it was necessary because we were concerned that if we did not ensure confidentiality then the respondents may report responses in a socially desirable manner about questions measuring PSRB, Psychological safety and leadership style as well, confidentiality was ensured to minimize the risk of social desirability bias, this is mentioned in detail in the “sample and procedures” section of the manuscript. Therefore if you are seeking for publically sharing the data then we must first seek respondents’ permission and ethical review board permission. We may public the data upon reasonable request from the editor. Institutional review board statement and Data availability and accessibility statements are given at the end of the manuscript before references. Thanks

We hope that the revision is satisfactory to you. We have provided our detailed responses and specific changes to the editor Comments point-by-point in the revised manuscript. 

Thank you so much.

Best regards 

Naqib Ullah Khan & Co-authors 

Previous response to reviewers is given bellow: 

Table 1 Response to the reviewer comments 

1 1. In the Introduction the authors describe positive rule-breaking as always unselfish (l.57). However, this is not necessarily the case. Positive rule-breaking only implies a positive consequence for others or e.g., an organization. In turn, this does not exclude profits for the individual. Actually, in most cases a positive consequence for the organization might also lead to positive consequences for the individual. For instance, the company increases revenues, so the individual gets a higher salary.

R Dear Sir, we have rewritten the introduction section and almost all the manuscript, the above mentioned sentence is modified and an effort is made to use proper words and sentences throughout the manuscript. Please look at first the file with tracked changes and then carefully review the file with untracked changes. Please focus on the file with untracked changes as all the changes made there are even not mentioned in the tracked file. Thanks 

2 2. There is a big overlap between the Introduction and Hypotheses development. The authors might consider to check for such overlaps in order to shorten both sections.

R Dear Sir, we have tried to remove the overlaps between both the Introduction and Hypotheses development sections, except this we also removed unnecessary and sentences from both the sections and other parts of the manuscript focusing to bring conciseness in the text. Please have a look on the untracked file of the manuscript and direct us for further actions if required. Thanks

3 3. Please provide a power analysis for your sample.

R Dear Sir, power analysis is normally conducted before the data collection and in studies where humans and animals’ blood samples or other such kind of samples are taken, where life threatening is risk likely. The main purpose underlying power analysis is to help the researcher to determine the smallest sample size that suitable to detect the effect of a given test at the desired level of significance, and to rely on smallest sample size to save time and resources. Our sample size is more than the required, according to the formula of number of items of the study measures multiplied by 10. In the literature that I have read and used, no one have used and prefers to express statistics of power analysis, it will extend the text of our study. But if you still emphasize then let me know, I will add a paragraph of power analysis before the measure section in the methodology, we can include statistics and explanation of power analysis on your emphasis in the next round of review. Thanks

4 4. It is not really clear for me how the exact procedure of the study looks. The authors should provide more detailed information here, e.g., in which order did participants fill out the questionnaires.

R Dear Sir, the sample and procedures section is rewritten completely and the order of taking responses from sampled participants is further clarified, please have a look and direct us if further editing or refining is required. Thanks 

5 5. The authors might consider to upload anonymized raw data to the open science framework (or a comparable provider) in order to enhance transparency.

R Dear Sir, please look at the changes made in the manuscript, if the changes are accepted, then please let us know to upload the raw data in the open science framework. Thanks

6 6. I am not an expert regarding this kind of analyses, so I will not go into detail regarding the data analysis. However, there are some formal issues. E.g., the value for beta is missing in line 415. Also regarding the statistics in Table 2: p = .00 is very rarely the correct expression. Usually, you indicate p-values to three decimal places and would state p < .001.

R Dear Sir, Beta value is added at line 350-351 and the p-values in table 2 are expressed in three decimal places. Please have a look. Thanks

7 7. In general, the whole manuscript requires further attention regarding details like spelling and punctuation. There are several mistakes (e.g., l. 376,386,421,610-612 to just name a few).

R Dear Sir, we have tried to remove the spelling and punctuation related errors throughout the manuscript. Please look at the manuscript. Thanks 

8 8. The same applies to the reference section. Several references are not cited correctly (e.g., missing page numbers, wrong capitalization,…). Doi numbers are missing for all references as well.

R Dear Sir, we have tried to present all the references correctly, page numbers and capitalizations are corrected, and DOIs are added using endnote. Thanks

---

## [Decision Letter · Decision Letter 1]

24 Feb 2023

Servant Leadership and Employee Prosocial Rule-Breaking: The Underlying Effects of Psychological Safety and Compassion at Work

PONE-D-22-15473R1

Dear Dr. Khan,

We’re pleased to inform you that your manuscript has been judged scientifically suitable for publication and will be formally accepted for publication once it meets all outstanding technical requirements.

Final editor comments: The referee was overal satisfied with the revision and provided some points (typos, etc.) to address in a final version of the manuscript. I suggest addressing all the remaining points before publication, so further rounds will not be necessary.

Kind regards,

Michael B. Steinborn, PhD

Section Editor

PLOS ONE

Additional Editor Comments (optional):

Reviewers' comments:

Reviewer's Responses to Questions

**Comments to the Author**

1. If the authors have adequately addressed your comments raised in a previous round of review and you feel that this manuscript is now acceptable for publication, you may indicate that here to bypass the “Comments to the Author” section, enter your conflict of interest statement in the “Confidential to Editor” section, and submit your "Accept" recommendation.

Reviewer #1: All comments have been addressed

2. Is the manuscript technically sound, and do the data support the conclusions?

Reviewer #1: Yes

3. Has the statistical analysis been performed appropriately and rigorously? 

Reviewer #1: I Don't Know

4. Have the authors made all data underlying the findings in their manuscript fully available?

Reviewer #1: No

5. Is the manuscript presented in an intelligible fashion and written in standard English?

Reviewer #1: Yes

6. Review Comments to the Author

Reviewer #1: I thank the authors for carefully addressing the comments from my prior review. Thereby, the quality of the manuscript has been improved significantly and I can recommend publication of the article.

However, there are still a few, minor points, the author should address before publication. This regards a few cases of spelling mistakes and in Table 2 (p.18) some p values are still reported as .000 instead of >.001.

7. PLOS authors have the option to publish the peer review history of their article (what does this mean?). If published, this will include your full peer review and any attached files.

Reviewer #1: No

---

## [Editor Report · Acceptance letter]

14 Apr 2023

PONE-D-22-15473R1 

Servant Leadership and Employee Prosocial Rule-Breaking: The Underlying Effects of Psychological Safety and Compassion at Work 

Dear Dr. Khan:

I'm pleased to inform you that your manuscript has been deemed suitable for publication in PLOS ONE. Congratulations! Your manuscript is now with our production department. 

Kind regards, 

on behalf of

Dr. Michael B. Steinborn 

Section Editor

PLOS ONE